# Stress and Well-Being of Greek Primary School Educators: A Cross-Sectional Study

**DOI:** 10.3390/ijerph20075390

**Published:** 2023-04-04

**Authors:** Dimitrios G. Zagkas, George P. Chrousos, Flora Bacopoulou, Christina Kanaka-Gantenbein, Dimitrios Vlachakis, Ioanna Tzelepi, Christina Darviri

**Affiliations:** 1Postgraduate Course of Stress Management and Health Promotion, Medical School, National and Kapodistrian University of Athens, 11527 Athens, Greecechridarviri@yahoo.com (C.D.); 2Center for Adolescent Medicine and Adolescent Health Care, First Department of Pediatrics, Medical School, National and Kapodistrian University of Athens, 11527 Athens, Greece; 3University Research Institute of Maternal and Child Health and Precision Medicine, Medical School, National and Kapodistrian University of Athens, 11527 Athens, Greece; 4Aghia Sophia Children’s Hospital, National and Kapodistrian University of Athens, 11527 Athens, Greece; 5Laboratory of Genetics, Department of Biotechnology, School of Applied Biology and Biotechnology, Agricultural University of Athens, 11855 Athens, Greece

**Keywords:** teachers’ well-being, perceived stress, quality of sleep

## Abstract

The teaching profession has always been challenging, while for various reasons the magnitude of observed stress in teachers has been continually growing over time. This study was conducted to demonstrate the relevance of stress in this professional group and to generate evidence for the benefit of primary school teachers and, indirectly, their pupils. To this end, we examined a large number of school teachers in a descriptive cross-sectional study. The survey comprised 786 primary school instructors aged 21 to 65 years, 646 women (82.2%) and 140 males (17.8%), and was performed from March to October 2022. Participants were asked about their gender, age, marital status, place of domicile, satisfaction with their income, whether their income met their needs, number of children, whether they cared for a person with a disability, work experience, alcohol use, eating patterns, and their height and weight for computation of their Body Mass Index (BMI). The survey included the Teacher Subjective Well-being Questionnaire (TSWQ), the Perceived Stress Scale (PSS), the Healthy Lifestyle and Personal Control Questionnaire (HLPCQ), and the Pittsburgh Sleep Quality Index (PSQI). The results showed that there were significant differences between the two sexes in age, marital status, work experience, smoking, alcohol use, and eating breakfast. Furthermore, there were significant differences between the two sexes in BMI, PSS Total, Dietary Health Choice, Harm Avoidance and Total HLPCQ. The variance of PSS Total was predicted by Sex, Teacher Efficacy, Total PSQI, Dietary Health Choice, organized physical exercise, social support and mental control, and Total HLPCQ. Between teacher efficacy, school connectedness, teacher well-being, organized physical exercise, social support and mental control, Total HLPCQ and PSS Total, the correlation coefficients were negative and significant at the <0.05 level. Between Total PSQI and PSS Total, the correlation coefficient was positive and significant at the <0.05 level. Between teacher efficacy, school connectedness and teacher well-being, organized physical exercise, social support and mental control, Total HLPCQ and Total PSQI, the correlation coefficients were negative and significant at the <0.05 level. In summary, we demonstrated that Greek primary school teachers experience significant stress, which is intertwined with their way of life, and reflected in significant decreases in their sense of well-being, quality of sleep, and overall life satisfaction, as well as in their standards of teaching.

## 1. Introduction

Teachers play a special and important role in children’s lives because, in addition to being facilitators of their learning, they are key pillars of a child’s socialisation, supporting and helping students to reach their highest potential and to develop into responsible citizens later in life. Teaching has always been a difficult process for educators, but under the current conditions, the level of stress has been constantly increasing. Indeed, studies conducted earlier reported that teachers experience higher stress than other professional groups—and considerably greater than the wider general population—with stress in teachers being a cross-cultural phenomenon [1,2].

The COVID-19 pandemic, from the spring of 2020 and during the entire academic year 2020–2021, caused a variety of changes in the education system to cope with the new demands and challenges created [3]. The changes in teaching with the modification of traditional methods of knowledge sharing and the introduction of entirely new technologies—with the peculiarities and problems associated with them—certainly affected the mental health of teachers and should be investigated [4,5].

In the US, 44% of primary school teachers say they “always” or “very often” experience burnout, the highest of all other professions in the US [6]. A figure of 35%, the second highest level of burnout, includes college and university workers [6]. Therefore, workers at almost all levels of education constitute one of the most stressed occupational groups in the US [6]. The process of teaching and, therefore, the academic performance of students, is negatively affected by teacher stress, with teachers who experience high levels of stress being encouraged to resign, causing a serious untoward impact on the school and the wider community including students, their families, as well as educators and their families; a phenomenon that is particularly common in the US [6].

The effects of occupational stress on teachers appear in various forms which can be: burnout in all of its forms (mental, emotional, physical)—as it is a chronic reaction to intense occupational stress—various psychosomatic symptoms, and depression [7]. These affect the quality of life of the teacher in his/her daily life, resulting in the simultaneous disruption of his/her work in the classroom, interpersonal relationships with students and colleagues, and an associated decrease in the degree of professional satisfaction [7]. A negative emotional reaction of teachers exposed to occupational stress is variously expressed as feelings of frustration, anxiety, anger, depression, despair, etc., and is influenced by the individual’s perception of a situation that may be potentially unpleasant or threatening, and by the inherent activation of specific defence mechanisms [8,9]. Moreover, chronic work stress in teachers can be a cause of an individual’s overall poor health and can even lead to emotional or physical injury [10].

A broader interpretation defines stress as a process of emotional and physical reactions, due to new or increased stressors on the individual, which exceed the individual’s coping capabilities. Therefore, when the individual perceives that his or her available resources, existing capabilities and needs are inferior to the demands of the job, a stress reaction is triggered. The individual’s stress level depends on his/her subjective perception of occupational challenges, coping strategies, anticipating any future demands and, finally, the individual’s degree of preparedness and ability to cope with the pressures in the workplace [11].

Important elements in the understanding and a more thorough study of teachers’ occupational stress are the identification of its sources. These can be divided into four main categories: (a) the school leadership and the cultural climate of its culture, (b) the individual job demands, (c) the support for decision-making and the autonomy allowed, and (d) the teacher’s emotional and social competence combined with personal resources. The stress sources, or “stressors”, mentioned above do not necessarily have to be in the order presented; they may coexist and it is possible that even one of them may be sufficient or of sufficient magnitude to cause clinically important stress [10].

The above categories should be analysed separately, in order to obtain a better understanding of them. The first category, school leadership structure, includes the quality of leadership, and the general climate and culture within the school [12]. The term leadership refers to the person who runs the school unit, the climate and culture refer to the relationships between employees (collegiality and cooperation), and the individual support provided [12,13]. The coexistence of the first three elements has been linked to high levels of satisfaction among long-term teachers and to the intention of new teachers to continue teaching [12,13]. At the relational level, low satisfaction in interactions with colleagues, students and supervisors has been observed to increase the intensity of stress [14], lead to lower levels of job satisfaction [15], and negatively affect feelings of loyalty towards the students [16]. The leadership level suffers in schools located in areas with high poverty levels, that are staffed with young inexperienced teachers, and that do not set high educational goals [17].

The second source of stress mentioned is the demands of the teacher’s job and the daily challenges. Daily student behaviour problems and their management, cooperation with parents and the demands they make, are responsible for teachers’ chronic stress problems and their vulnerability to depression [18]. High educational goals, either at the school environment or at the district level, negatively affect the pace and content of instruction and result in teachers feeling limited in their control of the didactic process [18].

The third source of stress concerns the supportive context and the degree of autonomy in decision-making. Autonomy and collaboration among teachers promoted through opportunities created by the leadership of the school empowers teachers and results in higher satisfaction levels [19]. However, the percentage of autonomy reported by teachers has declined, with the index of reported low autonomy rising from 18% in 2004 to 26% in 2012 [19]. Their involvement and support for engagement in decisions in the school in which they work, focus on realistic goals, supportive peer networks and sufficient level of control positively contribute to teachers’ health through reducing the effects of stress [20].

Last, but not least, are the personal resources and the social-emotional capacity of teachers. The combination of low social and emotional abilities and management skills with high demands on the teaching process and, hence, increased stress, lead to mental and physical deterioration and poor performance [21]. In addition, student outcomes are affected to a significant extent by the level of a teacher’s competencies and skills [22]. Inadequate training to develop competencies and skills due to lack of opportunities, concomitant with stress, create deficiencies resulting in negative impacts on classroom well-being and performance. However, teachers who are able to manage their emotions provide better support to students, when necessary, as well as positive reinforcement [23,24]. It is noted that high feelings of job satisfaction and personal sense of accomplishment are associated with enhanced social and emotional competence of teachers [23,24].

The area of teachers’ well-being has not been sufficiently examined, as studies measuring subjective well-being are limited to the professional group of teachers. Teaching directly depends on teachers’ well-being, is linked to teaching performance, and is a key prerequisite for education. Teachers should feel satisfied with themselves to be able to effectively function in the teaching–learning process, to be able to provide favourable conditions for students in the educational context and to make the most of their ability to create appropriate conditions conducive to learning and the well-being of students in the educational context.

The need to measure teachers’ well-being is considered an important element in understanding parameters such as the teacher’s attachment to the school, self-efficacy in teaching and, ultimately, teachers’ stress, elements that threaten their emotional and mental health and lead to burnout syndrome.

These background remarks highlight the importance of research dealing with the stress of this professional group critical for society. Scientific data are useful in the understanding of the adverse aspects of the working conditions of teachers and in devising their improvement for the benefit of primary school teachers and, hence, their students.

## 2. Materials and Methods

### 2.1. Participants and Procedures

The present study is a cross-sectional epidemiologic study. The aim of such studies is to investigate the relations between many relevant variables without taking into account their chronological order. These studies allow an estimate of the prevalence of a parameter and may provide information on the aetiology, albeit to a limited extent. They are quantitative, with a statistical basis; therefore, quantitative methodologies were used to draw conclusions about perceived stress, teacher well-being, sleep quality, healthy lifestyle and personal control.

The primary aim of this study was to investigate stress and well-being of teachers. Well-being will be examined through the validation of the TSWQ questionnaire among Greek teachers.

The secondary aims of the study were to examine gender differences and to search for predictors and correlations that may influence the complex issue of stress and well-being of teachers. It also aimed to extract data regarding healthy lifestyle adoption, sleep and demographic characteristics representative of the sample.

The same questionnaires were used for all primary school teachers who participated in the study. The criterion for inclusion was primary school teachers of any age in Greece, while teachers of other educational levels were excluded.

The study was conducted online with the questionnaire available on Google Forms, using a survey management software aimed at the largest geographical and populational coverage in Greece, from March 2022 to October 2022. The sample size was calculated using Cochran’s formula and, due to limited data regarding the topic to begin with, we assumed that half of the teachers had problems with stress, sleep, and the other variables in the sample; this provided us with the maximum possible variability. As a result, *p* = 0.5. We desired 95% confidence and precision of at least 5% above or below. According to the normal tables, a 95% confidence level yields Z values of 1.96, so we had ((1.96)^2^ (0.5) (0.5))/(0.05)^2^ = 385; thus, a random sample of 385 teachers in our target population was deemed sufficient to provide the confidence levels we required. The effects description was removed. More specifically, to reach the volunteers, a communication effort was prepared with the Greek Teachers’ Federation to have the questionnaire distributed by the official teacher representatives. The response was positive towards the questionnaire and was met with particular interest, but, unfortunately, the distribution was not successful because it would have set a precedent with the administration of questionnaires for specific purposes (e.g., racial discrimination). The questionnaires were available through a web forum https://www.pde.gr/ accessed on 15 June 2022, which is a community for teachers to raise issues that are subsequently analysed. The questionnaires remained available for four months. In the same time period, emails were sent to every school in the country through a contact list available from official bodies at two periods of time, first from mid-June to mid-July, a period during when teachers were finished teaching, and the second, from mid-September to mid-October, when they returned to teaching.

### 2.2. Measures

Participants were asked about their gender, age, marital status, place of residence, their income satisfaction, if family income covered their needs, the number of children they had, if they cared for a person with disabilities, their work experience, if they consumed alcoholic beverages and, if so, with what frequency, the amount they ate and their schedule, if they ate breakfast, as well as their height and weight, with the aim of computing their BMI.

*Teacher Subjective Well-being Questionnaire (TSWQ)*: the TSWQ questionnaire was created by Renshaw and colleagues in 2015 to assess teachers’ subjective well-being and several psychological parameters. It consists of eight items, with the response to each question rated on a four-point scale (from 1 = almost never to 4 = almost always) and includes questions such as “I feel like I belong in this school” and “I have achieved a lot as a teacher”. The TSWQ questionnaire consists of two subscales: (1) Teaching Effectiveness, and (2) School Connectedness, with the scores of each scale used either independently or summed up to compose the overall measure of teacher well-being. The Teaching Effectiveness subscale consists of the addition of responses to questions 2, 4, 6, 8, while the School Connectedness subscale consists of 1, 3, 5, 7 [25].

The subjective well-being questionnaire can be used in combination with measures of negative psychological functions such as: teacher stress and teacher burnout [25]. An interesting theme that emerged in the results of the TSWQ questionnaire was that, whereas there was no difference between educational grades, there was a difference in responses between schools for the same educational grade [25]. Permission for the translation was requested by Tyler L. Renshaw, who developed the TSWQ. The translation of the questionnaire was undertaken by an expert committee in accordance with the principles of the World Health Organization [26]. Points that caused confusion during translation to Greek were identified during the test-pre-test of the research tool and corrections were made. The sample selection for the test-pre-test procedure emerged through the representativeness of the study population in terms of native language and age range and consisted of primary school teachers (35 women, 5 men). After these procedures, the Greek version of the questionnaire was finalized. All rights to the TSWQ questionnaire belong to the authors.

An online reference survey was administered to teachers from around the nation. The TSWQ, the Perceived Stress Scale (PSS), and the Healthy Lifestyle and Personal Control Questionnaire were all part of the study. A total of 786 people took part in the survey. According to our findings, the Greek version of the TSWQ questionnaire may be utilized in combination with measures of negative psychological functioning, such as teacher stress and burnout. The exploratory factor analysis (EFA) of the eight items explained 52.4% of the overall variation. The internal consistency of the TSWQ scale was good, with Cronbach’s alpha at 0.866, the Teaching Effectiveness subscale at 0.833, and the School Connectedness subscale at 0.880. Researchers and practitioners may use this measure to examine the well-being of elementary teachers.

*Perceived Stress Scale (PSS):* The scale consists of 14 items (PSS-14), developed originally in English [27], and consists of seven questions with positive items and seven questions with negative items. It was developed as a self-reporting questionnaire to assess the level of perceived stress over the past month based on responses using a 5-point Likert scale (0 = never, 1 = hardly ever, 2 = once in a while, 3 = often, 4 = very often). The calculation of the total score is obtained by inverting the answers to questions 4, 5, 6, 7, 9, 10 and 13 (in the following way: 0 = 4, 1 = 3, 2 = 2, 3 = 1, and 4 = 0) and adding all the answers [27]. The highest score is 56, with the highest score indicating a higher level of perceived stress [27].

In Greece, Alexia Katsarou et al. performed the translation, reliability and validity testing of the Greek version of the PSS-14 and found it a valid tool for the assessment of perceived stress in Greek adults, without providing a diagnosis, but rather defining it as a risk factor [28].

*Healthy Lifestyle and Personal Control Questionnaire (HLPCQ):* in the HLPCQ questionnaire, participants indicate the frequency of adoption of 26 lifestyle questions with a positive bias, on a Likert-type scale (1 = never or rarely, 2 = sometimes, 3 = often and 4 = always). It consists of 12 items related to Healthy food choices and Avoiding dietary harm, eight related to Daily Routine, two related to Organized Physical Activity, and four related to Social and Mental Balance. The subscales are characterized by good internal consistency and variance, while their scores, as well as the total score, have been significantly correlated with perceived stress and health status [29]. The HLPCQ questionnaire was formatted and validated in the Greek language by Darviri et al. [29].

*Pittsburgh Sleep Quality Index (PSQI):* the Greek version of the PSQI questionnaire was distributed to assess the sleep quality of the participants during thirty (30) days preceding the day of completion [30]. The PSQI questionnaire includes 19 self-reporting questions, grouped into seven components, which are: (a) subjective sense of sleep quality, (b) wake time, (c) latency, (d) duration, (e) habitual sleep productivity, (f) sleep medication use, and (g) daytime dysfunction [30]. There are an additional five questions to be completed by a close person, such as a spouse or parent, but these are not included in the final score. (These questions were not included in the present study [29]). The scores for each component use values ranging from 0 to 3, with 0 representing no sleep distress and 3 indicating severe sleep distress, with the total score taking values from 0 equating to high sleep quality to 21 representing low sleep quality [31]. A total score on the PSQI questionnaire with a value above five (>5) indicates poor sleep quality, while a score ≤ 5, represents a relatively or fairly good sleep quality [31]. It has a high specificity and sensitivity for distinguishing between “poor” and “good” sleepers [30,31]. The questionnaire was validated and tested for reliability in the Greek population by Kotronoulas et al. [30].

### 2.3. Ethical Considerations

The Committee of the Medical School of the National and Kapodistrian University of Athens approved the study protocol, which was in accordance with the Declaration of Helsinki (2013). In the first part of the Google form, the volunteers who participated in the study were duly informed about the nature, significance, implications, risks and confidentiality of the study, and were given the option to consent or refuse to participate.

### 2.4. Method of Data Analysis

Categorical variables were presented as frequencies and percentages, while quantitative variables were shown as median and interquartile range (IQR). The Kolmogorov–Smirnov test was performed for normality evaluation. The non-parametric Spearman correlation coefficient rho was used to investigate the existence of a linear relationship. The chi squared (χ2) test was used to correlate categorical variables and the nonparametric Mann–Whitney U test was used to correlate quantitative and categorical variables. Linear regression models were used to estimate differences in participant characteristics and instruments used as predictors of teacher stress. Logistic regression models were used to evaluate various participants’ characteristics as determinants of which category of perceived stress (Low/Moderate vs. High) each individual was likely to belong in. The level of statistical significance was set at *p* < 0.05. Analyses were conducted using SPSS 26.0 for Windows.

## 3. Results

The demographic characteristics of the participants are presented in Table 1. A total of 786 subjects participated in the study, of which 82.2% were female and the rest male; 43.3% of the subjects belonged in the age group 36–50 years; there was a significant statistical difference between the two sexes and age groups (*p* < 0.001). Regarding BMI categories, most of the participants were in the “normal weight” category (57.9%); however, there were significant percentages of teachers in the “overweight” and “obese” categories (28.8% and 11.2%, respectively). There was a statistically significant difference in the BMI and gender categories, with 51.4% of males in the overweight and 64.1% of the females in the normal weight status (*p* < 0.001). Most participants were in the married category (66.9%), with men being married at a higher proportion than women (76.4% vs. 64.9%, *p* = 0.048). Regarding work experience, 61.3% of participants had >15 years of experience, with men having a higher proportion in this category than women (*p* = 0.028).

In terms of smoking, 66% of the sample were non-smokers; there was a significant difference by gender (*p* = 0.004). More specifically, more men reported being ex-smokers than women. On alcohol consumption quantity and frequency, the response was ‘no’ and ‘rarely’ in 30.5% and 27.7% of the subjects, respectively, with one in five people stating that they consumed alcohol 1–2 times per week. There was a significant difference between the genders (*p* = 0.031), with men consuming alcohol more frequently than women. When asked if they ate breakfast, 55.3% of responses were “always”, with a higher percentage of women than men (*p* = 0.040).

In the categorization of “perceived stress”, most of the respondents said they belonged to the “moderate” category (72.6%), while 8.7% said they were under “high” stress. In further categorization, 91.3% were in the “low/moderate” category. There was a significant difference between the two sexes with respect to perceived stress in both categorizations (*p* = 0.045 and *p* = 0.0028, respectively).

The questionnaire scores used in the study are presented in Table 2. Mean values (standard deviations), medians (interquartile ranges) for the overall questionnaires and their subscales are presented. To find statistically significant differences between the sexes, descriptive measures are presented separately.

Men had a higher BMI than women (*p* < 0.001). In the final score of the perceived stress scale, the women scored higher than men (*p* = 0.011). In the Healthy Lifestyle questionnaire, significant differences were found in two subscales and in the total score. More specifically, women scored higher on the subscale of having healthy eating episodes (*p* < 0.001) and avoiding harmful eating episodes (*p* < 0.001). In the total scale score, women scored higher than men (*p* = 0.005).

The correlations between the quantitative variables are presented in Table 3. We observed a negative correlation between BMI and the subscales of healthy eating outcomes, avoidance of harmful food, organized physical activity, social support and mental health, and in the final score of the HPLPQ scale (r = −0.219, r = −0.239, r = −0.121, r = −0.122, r = −0.219, respectively). Teacher Effectiveness was significantly and positively correlated with the subscale of teacher’s attachment to school, sense of well-being, healthy food choices, daily routine and the final score of HPLPQ scale (r = 0.497, r = 0.859, r = 0.093, r= 0.119, r= 0.135, respectively), while there was a significant negative correlation with the final score of sleep (PSQI) and perceived stress (PSS) (r = −0.122, r = −0.186, respectively). The final scale of Teacher Well-being was positively correlated with the subscales of healthy food choices, organized physical activity and the final score of the HPLPQ scale (r = 0.128, r = 0.104, r = 0.156, respectively), while it was significantly negatively correlated with the final scores of the PSS and PSQI scales (r = −0.089, r = −0.185, respectively). Connectivity to school was significantly negatively correlated with the final score of the PSS scale (r = −0.129) and also positively correlated with the TSWQ scale, healthy food choices, organized physical activity, social support and mental health and with the final score of the HPLPQ scale (r = 0.861, r = 0.140, r = 0.175, r = 0.115, r = 0.145, respectively).

The correlation of the PSQI Final Score was positively significantly correlated with the final score of the PSS scale (r = 0.475). In contrast, the sleep scale was negatively significantly associated with healthy food choices, avoidance of harmful outcomes, organized physical activity, social support and mental health and with the final score of the HPLPQ scale (r = −0.075, r = −0.090, r = −0.191, r = −0.208, r = −0.232). The final score of the PSS scale showed a positive statistically significant correlation with all subscales and with the final score of the HPLPQ scale (r = −0.141, r = −0.190, r = −0.096, r = −0.184, r = −0.232, r = −0.313, respectively).

The subscales of the HPLPQ questionnaire were significantly positively correlated with the final HPLPQ scale score (r = 0.765, r = 0.675, r = 0.367, r = 0.367, r = 0.268, r = 0.511, respectively).

Table 4 presents the results of the linear regression analyses for the PSS questionnaire of perceived stress in relation to the subscales of the well-being questionnaire, the final score of the PSQI and HPLPQ scales and demographic characteristics. Regression analysis was applied to examine whether the independent variables gender, age categories (36–50 years, 51–64 years), work experience categories (6–10 years, 11–15 years, >15 years), the Teacher Effectiveness subscale, the School Connectivity subscale, PSQI and HPLPQ scales were predictors of the PSS scale. The results revealed that the multiple R of the regression analysis was 0.53, which is statistically significantly different from zero, F (10, 775) = 30,512, *p* < 0.001. Overall, all 12 independent variables explained 27.3% of the variance in the PSS Total. We had 12 coefficients in the total, 10 for the independent variables and two for the constants. In terms of gender, women had a higher score by 1.805 points (*p* = 0.010) than men. The final PSQI scale score was significantly positively associated with higher levels of perceived stress by 1.344 points (*p* < 0.001). The teacher effectiveness subscale and the final score of the HPLPQ scale were significantly negatively associated with lower levels of perceived stress by −1.244 points (*p* = 0.008) and −0.150 points (*p* < 0.001), respectively. In contrast, the variables age categories, work experience categories, and school attachment subscale had no statistically significant contribution to stress levels.

The results of the linear regression analyses for the TSWQ questionnaire of well-being in relation to the final score of the PSS, PSQI and HPLPQ scales, and demographic characteristics are presented in Table 5. Regression analysis was applied to examine whether the independent variables age categories (36–50 years, 51–64 years), work experience categories (6–10 years, 11–15 years, >15 years), PSQI, HPLPQ and PSS scales were predictors of the TSWQ scale. The results revealed that the multiple R of the regression analysis was 0.27, which is statistically significantly different from zero, F (8, 777) = 7.908, *p* < 0.001. Overall, all 10 independent variables explained 6.7% of the variance in the PSS Total. We had 10 coefficients in the total, eight for the independent variables and two for the constants. The final HPLPQ scale score was significantly positively associated with higher levels of perceived stress by 0.004 points (*p* = 0.021). In addition, the category of work experience (>15 years) was significantly positively associated with higher levels of perceived stress by 0.261 points (*p* = 0.002). The final score of the PSS scale and the category of age (36–50) were significantly negatively associated with lower levels of perceived stress by −0.009 points (*p* = 0.002) and −0.196 points (*p* = 0.006), respectively. In contrast, PSQI scale, age category (36–50) and categories of work experience (6–10 and 11–15) had no statistically significant contribution to stress levels.

Logistic regression revealed that the final score of the HLPLPQ scale and age category (51–65 years) rather decreased the probability of belonging to the High PSS category, while work experience category (6–10) and Total PSQI increased the probability of belonging to the High PSS category. More specifically, Table 6 presents the results of logistic regression analysis models of various participant characteristics as determinants of which category of perceived stress (Low/Moderate vs. High) each individual was likely to belong to. Individuals belonging to the age category (51–65) had decreased odds of being in the high stress category (OR = 0.252, *p* = 0.016) compared to individuals in categories (21–35) and (36–50). Furthermore, individuals belonging to the category of work experience (6–10 years) had a higher odds ratio of belonging to the high stress category (OR = 3.758, *p*= 0.011). In contrast, individuals scoring higher on the healthy lifestyle scale (HLPLPQ scale) had a lower odds ratio of belonging to the high stress category (OR = 0.967, *p* = 0.005). In addition, a higher score on the sleep scale increased the likelihood of the individual being in the high stress category (OR = 1.305, *p* = <0.001).

## 4. Discussion

Summarizing the results of this study, high rates of overweight/obesity, weekly alcohol consumption, and smoking were observed among teachers. The results of the current research are consistent with those of prior studies on high obesity rates, such as one in which the following findings were reported: 85 percent of instructors had abdominal obesity, 56 percent had obesity as measured by the proportion of fat mass, and 33 percent had hypertension [32]. According to the findings of a 2015 study performed by the Substance Abuse and Mental Health Services Administration (SAMHSA), 4.7% of teachers and workers in the field of education as a whole acknowledged consuming alcohol excessively in the previous month. This data is concerning. This is a substantial rise from the 3.7% rate in 2008 [33].

The hazardous habit of smoking has been proven to impact the health of both instructors and teenagers. Smoking by teachers during school hours is connected with adolescent smoking [34]. Furthermore, smoking by male educators tends to influence boys’ smoking, especially during their initial two years of secondary school [35].

In addition, the majority of participants reported being in the moderate stress category, while more than half of them experiencing sleep problems. Thus, the evidence from the study is particularly worrying for teachers’ well-being and health in general. On the other hand, analysing the correlation results, we observed that increased BMI scores were associated with decreases in habits associated with a healthy lifestyle, while, on the contrary, increased teachers’ effectiveness led to a decrease in the final scores of both the PSS and PSQI scales and an increase in the degree of adoption of a healthier lifestyle. Thus, as expected, an increase in well-being is associated with a decrease in the total score on the PSS and PSQI scales and teacher effectiveness. Regarding sleep, we observed that as the final score on the scale studying sleep increases, the final score on perceived stress also increases. Studies of professional groups during the COVID-19 era revealed that during the early months of the pandemic’s emergence, poor sleep quality and reported stress were prevalent among healthcare personnel. Two years into the pandemic, reported stress decreased as sleep quality deteriorated [36]. High stress was related with severe sleep length and quality disruptions [36]. Stress levels were also connected with the daytime effects of sleep disturbances [36]. The relationship between stress and sleep may be an essential mechanism mediating the correlation between stress and cardiovascular disease [37].

Therefore, an increase in the score on the sleep scale negatively affects the subscales and the overall healthy lifestyle and personal control scales. Finally, an increase in the final score of the PSS scale led to a decrease in the adoption of healthy habits. Sleep quality is one of the five elements deemed significant for assessing healthy sleep, which is defined as a multimodal pattern of sleep–wake that is responsive to individual, societal, and environmental needs and offers physical and mental well-being [38]. As a fundamental human need, sleep is one of the most important new topics, as there is substantial evidence that sleep deficiency and sleep disturbances alter metabolic and inflammatory processes, with far-reaching adverse health effects [39].

Modern public health is based on health promotion and disease prevention. The Ottawa Charter for Health Promotion, often known as the Ottawa Charter, was signed in 1986 and defines the fundamental aspects of health promotion [40]. It states that the goal of health promotion is achievable so that individuals gain control over and improve their health. Consequently, we must identify and promote measures that enable individuals to maintain and improve their health [40]. To this end, we need to provide useful information about the behaviours and conditions that support health, as well as incentives and support for the acquisition of life skills that help apply this knowledge to real life. According to the Ottawa Charter, health promotion is just as (or more) important for achieving health as other health protection services [40].

Health promotion is a modern area of public health that focuses on individual behaviour as well as the general development of competence and health. Professional health promotion activities are carried out both within and outside the health sector. Effective health promotion programs need local action, as well as institutional responsibility for their initiation and implementation [40]. The World Health Organization’s “Best Buy” list summarizes the most cost-effective and feasible measures (for example, bans on alcohol advertising or increasing public knowledge about nutrition) that allow nations of all economic levels to reduce the burden of chronic non-communicable diseases [41]. Alongside developments in population-based research, there is growing scientific evidence of the health benefits of health promotion in health and the potential for disease prevention as a consequence of healthy lifestyles. These findings can also be used at different levels of expenditure.

Epidemiological research, with its widespread development in different populations at different times, has contributed to the creation and identification of activities that promote the maintenance and improvement of health. The list is progressively becoming longer and more precise. With regards to health, the following stable behaviours—which include lifestyle issues—are the most beneficial areas in which to work: (a) moderate physical exercise [42]; (b) a healthy diet; (c) avoiding health risks, such as smoking, excessive alcohol consumption, overweight/obesity, and falls [43,44]; (d) avoiding stressors, enhancing cognitive activity, cultivating and improving emotional intelligence, consolidating and developing strong social bonds and removing social exclusions [45,46,47,48]; and (e) sexual health care to prevent sexually transmitted diseases [49].

Based on the findings of this study, we understand the need to support teachers, with the aim of reducing their stress and improve their quality of life, and at the same time their performance in the school context in which they are embedded. Several interventions have been proposed to support teachers. These include (a) individual-focused interventions, which are the most widespread, (b) interventions linking the school context and the individual, i.e., the approach to learning skills and social support for the teacher, and (c) changes in the organization of the educational context and the educational system in general [50,51]. The first category of interventions, and the most common approaches, are individual efforts, such as meditation, relaxation exercises, cognitive behavioural, and metacognitive approaches. The primary focus of the interventions is to develop and/or improve existing stress-coping skills, as well as setting specific goals to achieve [51]. Teachers that have participated in stress management programs themselves express positive opinions and point out that they observe benefits in their mental and physical health [51].

The introduction of mindfulness, in individual stress management programs, encourages teachers to focus on the present, manage attention with intention, and observe things in a non-judgmental way, combining compassion, patience, and kindness to the individual [52,53,54]. Learning skills in the context of mindfulness includes exercises to control breathing, scanning the body, creating and further cultivating positive emotions towards others and towards the individual and increasing the level of emotional awareness. In studies using mindfulness-based interventions or similar intervention models, it has been observed that participating teachers experience physical and psychological benefits, such as a reduction in work stress and burnout, while improving the quality of their teaching [55,56].

Of particular interest are findings of teachers who were practicing Transcendental Meditation [57]. In this study, post-intervention teachers showed increased empathy and forgiveness, personal growth, and experienced lower stress [57]. Studies that focus on physiological changes in teachers are limited, but the existing findings are encouraging, as mindfulness-type interventions can reduce cortisol concentrations and blood pressure levels [58,59,60] and improve sleep quality [61,62,63].

An intervention using a modern stress management technique based on a cognitive and social approach called “Pythagorean Self-Awareness Intervention (PSAI)”, has presented quite encouraging results in “stressed” people with obesity, insomnia, multiple sclerosis [64], mild cognitive impairment [65] and acne vulgaris [66]. In a recent study in adults with obesity, an eight-week PSAI intervention was conducted, and a reduction in perceived stress, stress biomarkers—such as serum and salivary cortisol concentrations—and BMI, as well as obesity-associated metabolic markers were observed [67]. Similarly, in a study in patients with mild cognitive impairment, a PSAI intervention was associated with a significant improvement in perceived stress, depressive feelings and self-efficacy [65]. In addition, individuals suffering from chronic insomnia showed significant improvement in response to PSAI [68].

The second category of interventions pertain to the school–individual interface, i.e., providing new skills and social support to the teacher. Generally, new teachers experience high stress due to the novelty of the professional demands. In recent years, orientation and mentoring programs have been created, which focus on the technical and social support of teachers entering the school environment for the first time or with little earlier experience [69,70]. In this category, programs are carried out along two axes: (a) skills learning through programs promoting well-being in the school context, and (b) emotional and social education focusing on students’ behaviour [70]. In the first axis, the aim is to change lifestyle and thus, reduce health risk behaviours for individuals and their partners [71]. There has been an increase of over 20% in such programs in schools in the US over the last 15 years; they focus on the areas of nutrition, body weight management, physical activity and health risk assessment. Interestingly, since 2000, stress management programs have declined, with only one in four schools offering them at this time [71]. For participating professionals, interventions that have included wellness programs, insurance incentives and improved administrative procedures are highly encouraging. There was a percentage reduction in indicators such as: 34.7% in systolic blood pressure, 38.6% in total cholesterol, 46.0% in BMI, and 65.6% in blood glucose concentrations [72]. The second axis aims to promote positive behaviour (PBIS) in the school. More specifically, these programs include coaching to improve the quality of relationships and interactions with students. The results have shown a positive impact on teacher feelings and behaviour and student performance, and have emphasized the importance of ongoing social and emotional training of teachers [73,74].

The third category of interventions concerns changes in the organisation of the educational framework and the educational system; the goal is a multi-level change in the educational system and culture [75]. The main intervention programs are the redesign of workplaces (e.g., working hours, workload), support through supervision, promotion of a participatory environment, health policies for teachers and those working in the school environment; in general, interventions—some more flexible and feasible and some more rigid—meet a high degree of difficulty due to coordination complications, political decisions, and reaction to cultural changes [76].

The majority of the interventions presented extensively on stress included wellness programs. In our study, results on indicators such as body mass index, sleep quality, perceived stress, and healthy lifestyle choices and personal control prompt us to create programs to promote health and prevent non-communicable and communicable diseases.

Nevertheless, this study has several limitations: as the current sample is not typical of the entire population at large, our findings should be considered with caution; it has a cross-sectional character and, hence, it is impossible to define causality; participation bias is possible and may have influenced the results. To solidify the conclusions and to study possible moderators of the link between stress, well-being, and quality of life, larger, prospective studies are required.

## 5. Conclusions

In summary, the results of this study are interesting and contribute to highlighting the important issue of teachers’ well-being. Stress experienced by teachers is a serious negative factor in their effectiveness, as those teachers who are well mentally and physically are able to achieve the best possible results for their pupils. Teachers should be empowered to develop better interpersonal skills, to improve themselves, and to experience a better quality of life. The state at the national, regional and local levels should empower school communities, and by extension the individuals working, cooperating and learning in the school environments.

## Figures and Tables

**Table 1 ijerph-20-05390-t001:** The sample’s sociodemographic and lifestyle characteristics (N = 786).

		Study Measurements		
	Total (N = 786)	Males	Females	*p*-Value
Gender N (%) - Women - Men	646 (82.2) 140 (17.8)	-	-	-
Age N (%) - 21–35 years - 36–50 years - 51–65 years	147 (18.7) 340 (43.3) 299 (38.0)	12 (8.6) 46 (32.9) 82 (58.6)	135 (20.9) 294 (45.5) 217 (33.6)	<0.001
BMI Categories N (%) - Underweight - Health Weight - Overweight - Obese	21 (2.7) 455 (57.9) 222 (28.2) 88 (11.2)	0 (0) 41 (29.3) 72 (51.4) 27 (19.3)	21 (3.3) 414 (64.1) 150 (23.2) 61 (9.4)	<0.001
Marital status N (%) - Married - Unmarried - Divorced - Widowed	526 (66.9) 208 (26.5) 45 (5.7) 7 (0.9)	107 (76.4) 28 (20.0) 5 (3.6) 0 (0)	419 (64.9) 180 (27.9) 40 (6.2) 7(1.1)	0.048
Children N (%) - Yes - No	515 (65.5) 271 (34.5)	99 (70.7) 41 (29.3)	416 (64.6) 230 (35.6)	0.148
Caring for person with special needs (%) - Yes - No	69 (8.8) 717 (91.2)	10 (7.1) 130 (92.9)	59 (9.1) 587 (90.9)	0.451
Education level N (%) - Bachelor - Master - PhD	396 (50.4) 375 (47.7) 15 (1.9)	67 (47.9) 68 (48.6) 5 (3.6)	329 (50.9) 307 (47.5) 10 (1.5)	0.258
Residence N (%) - City > 150.000 - City < 150.000 - Province	319 (40.6) 246 (31.3) 221 (28.1)	54 (38.6) 45 (32.1) 41 (29.3)	265 (41.0) 201 (31.1) 180 (27.9)	0.864
Work experience N (%) - 1–5 years - 6–10 years - 11–15 years - >15 years	106 (13.5) 72 (9.2) 126 (16) 482 (61.3)	11 (7.9) 10 (7.1) 18 (12.9) 101 (72.1)	95 (14.7) 62 (9.6) 108 (16.7) 381 (59.0)	0.028
Income satisfaction N (%) - Not at all - A bit - Average - A lot - Very much	152 (19.3) 179 (22.8) 388 (49.4) 62 (7.9) 5 (0.6)	26 (18.6) 39 (27.9) 68 (48.6) 7 (5.0) -	126 (19.5) 140 (21.7) 320 (49.5) 55 (8.5) 5 (0.8)	0.299
Family income covers your needs N (%) - Not at all - A bit - Average - A lot - Very much	57 (7.3) 169 (21.5) 455 (57.9) 96 (12.2) 9 (1,1)	7 (5.0) 37 (26.4) 86 (61.4) 9 (6.4) 1 (0.7)	50 (7.7) 132 (20.4) 369 (57.1) 87 (13.5) 8 (1.2)	0.077
Are you a smoker N (%) - Yes - No - Ex	177 (22.5) 519 (66.0) 90 (11.5)	25 (17.9) 88 (62.9) 27 (19.3)	152 (23.5) 431 (66.7) 63 (9.8)	0.004
Do you consume alcoholic beverages; if so. with what frequency? N (%) - No - Daily - 1–2 times per week - 3–5 times per week - 3–5 times per month - Rarely	240 (30.5) 10 (1.3) 150 (19.3) 31 (3.9) 135 (17.2) 218 (27.7)	33 (23.6) 4 (2.9) 33 (23.6) 10 (7.1) 24 (17.1) 36 (25.7)	207 (32.0) 6 (0.9) 119 (18.4) 21 (3.3) 111 (17.2) 182 (28.2)	0.031
Do you think you eat? N (%) - A lot - Little - Regularly	138 (17.6) 51 (6.5) 597 (76.0)	34 (24.3) 7 (5.0) 99 (70.7)	104 (16.1) 44 (6.8) 498 (77.1)	0.061
Do you think you eat? N (%) - Fast - Slow - Regularly	349 (44.4) 80 (10.2) 357 (45.4)	71 (50.7) 11 (7.9) 58 (41.4)	278(43.0) 69(10.7) 299(46.3)	0.220
Do you eat breakfast? N (%) - Always - Sometimes - Rarely - Never	435 (55.3) 192 (24.4) 99 (12.6) 60 (7.6)	64 (45.7) 41 (29.3) 25 (17.9) 10 (7.1)	371 (57.4) 151 (23.4) 74 (11.5) 50 (7.7)	0.040
PSS Categories N (%) - Low - Medium - High	147 (18.7) 571 (72.6) 68 (8.7)	31 (22.1) 104 (74.3) 5 (3.6)	116 (18.0) 467 (72.3) 63 (9.8)	0.045
PSS Categories N (%) - Low/medium - High	718 (91.3) 68 (8.7)	135 (96.4) 5 (3.5)	583 (90.2) 63 (9.8)	0.028
PSQI Categories N (%) - Non-significant sleep disturbance - Significant sleep disturbance	364 (46.3) 422 (53.7)	72 (51.4) 68 (48.6)	292 (45.2) 354 (54.8)	0.213

**Table 2 ijerph-20-05390-t002:** The sample’s study measurements.

		Study Measurements		
Categories	Total (N = 786)	Males Median (IQR) (Q1–Q3)	Females Median (IQR) (Q1–Q3)	*p*-Value
BMI	23.83 (5.49) (21.45–26.94)	26.80 (4.51) (24.90–29.40)	23.19 (4.95) (21.18–26.13)	<0.001
Teacher Efficacy	3.00 (1.00) (2.50–3.50)	3.00 (1.00) (2.50–3.50)	3.00 (1.00) (2.50–3.50)	0.813
School connectedness	3.00 (1.00) (2.50–3.50)	3.00 (1.00) (2.50–3.50)	3.00 (1.00) (2.50–3.50)	0.837
Teacher Well-Being	3.00 (1.00) (2.63–3.50)	3.00 (1.00) (2.63–3.50)	3.00 (1.00) (2.63–3.50)	0.953
Total PSQI	6.00 (4.00) (4.00–8.00)	5.00 (3.00) (4.00–7.00)	6.00 (4.00) (4.00–8.00)	0.081
PSS Total	27.00 (11.00) (21.00–32.00)	25.00 (11.00) (19.25–30.00)	27.00 (11.00) (21.00–32.00)	0.011
Dietary Health choice	17.00 (6.00) (14.00–20.00)	16.00 (5.00) (13.00–18.00)	17.00 (5.00) (15.00–20.00)	<0.001
Dietary Harm Avoidance	11.00 (4.00) (9.00–13.00)	10.00 (4.00) (8.00–12.00)	11.00 (4.00) (9.00–13.00)	<0.001
Daily Routine	9.00 (18.00) (4.00–22.00)	10.00 (17.00) (5.00–22.00)	8.00 (17.00) (4.00–21.00)	0.439
Organized Physical Health	8.00 (9.00) (4.00–13.00)	8.00 (9.00) (4.00–13.00)	8.00 (9.00) (4.00–13.00)	0.834
Social Support and Mental Health	15.00 (9.00) (12.00–21.00)	15.00 (7.00) (12.00–19.00)	16.00 (9.00) (12.00–21.00)	0.149
Total HPLPQ	66.00 (16.00) (57.00–73.00)	64.00 (15.00) (55.00–70.00)	66.00 (17.00) (57.00–74.00)	0.005

**Table 3 ijerph-20-05390-t003:** Correlations between quantitative variables.

	BMI	Teacher Efficacy	School Connectedness	Teacher Well-Being	Total PSQI	Total PSS	Dietary Healthy Choice	Dietary Harm Avoidance	Daily Routine	Organized Physical Exercise	Social Support & Mental Control	Total HPLPQ
BMI	1.000	0.070 *	0.019	0.056	−0.002	0.003	−0.219 **	−0.239 **	−0.012	−0.121 **	−0.122 **	−0.218 **
Teacher Efficacy		1.000	0.497 **	0.859 **	−0.122 **	−0.186 **	0.093 **	0.048	0.119 **	0.015	−0.016	0.135 **
School Connectedness			1.000	0.861 **	−0.028	−0.129 **	0.140 **	0.062	−0.056	0.175 **	0.115 **	0.145 **
Teacher well-being				1.000	−0.089 *	−0.185 **	0.128 **	0.055	0.040	0.104 **	0.054	0.156 **
Total PSQI					1.000	0.475 **	−0.075 *	−0.090 *	−0.046	−0.191 **	−0.208 **	−0.232 **
Total PSS						1.000	−0.141 **	−0.190 **	−0.096 **	−0.184 **	−0.232 **	−0.313 **
Dietary Healthy Choice							1.000	0.520 **	0.230 **	0.137 **	0.277 **	0.765 **
Dietary Harm Avoidance								1.000	0.214 **	0.102 **	0.233 **	0.675 **
Daily Routine									1.000	−0.660 **	−0.449 **	0.367 **
Organized Physical Exercise										1.000	0.700 **	0.268 **
Social Support & Mental Control											1.000	0.511 **
Total HPLPQ												1.000

* Correlation is significant at the 0.05 level (2-tailed). ** Correlation is significant at the 0.01 level (2-tailed).

**Table 4 ijerph-20-05390-t004:** Results from multivariate linear regression analyses that evaluated various participants’ characteristics as determinants of PSS.

Coefficients ^a^			
	Unstandardized B	Std. Error	Sig
Sex	1.805	0.701	0.010
Age Categories (Ref: 21–35 y) 36–50 y	0.629	0.974	0.519
50–64 y	−0.197	1.134	0.862
Work experience (Ref: 1–5 y) 6–10 y	−0.224	1.127	0.084
11–15 y	−0.114	1.109	0.918
>15 y	−0.553	1.126	0.623
Teacher Efficacy	−1.244	0.467	0.008
School Connectedness	−0.504	0.452	0.265
Total PSQI	1.344	0.110	<0.001
TotalHPLPQ	−0.150	0.023	<0.001

^a^ Dependent Variable: PSS_TOTAL. y: years.

**Table 5 ijerph-20-05390-t005:** Results from multivariate linear regression analyses that evaluated various participants’ characteristics as determinants of TSWQ.

Coefficients ^a^			
	Unstandardized B	Std. Error	Std. Error
Age Categories (Ref: 21–35 y) 36–50 y	−0.196	0.071	0.006
50–64 y	−0.130	0.082	0.114
Work experience (Ref: 1–5 y) 6–10 y	0.020	0.083	0.805
11–15 y	0.107	0.081	0.189
>15 y	0.261	0.082	0.002
Total PSS	−0.009	0.003	0.002
Total PSQI	0.000	0.009	0.975
Total HPLPQ	0.004	0.002	0.021

^a^ Dependent Variable: TSWQ_scale. y: years.

**Table 6 ijerph-20-05390-t006:** Results from multivariate logistic analysis that evaluated various participants’ characteristics as determinants of PSS.

	Exp (B)	S.E.	Sig.
Sex	2.629	0.505	0.055
Age Categories (Ref: 21–35 y) Age (36–50 y)	0.692	0.436	0.020 0.398
Age (51–65 y)	0.252	0.570	0.016
Work experience (Ref: 1–5 y) 6–10 y	3.758	0.518	0.059 0.011
11–15 y	1.892	0.547	0.244
>15 y	3.003	0.578	0.057
Teacher Efficacy	0.847	0.240	0.490
School Connectedness	0.758	0.232	0.233
Total PSQI	1.305	0.056	<0.001
Total HPLPQ	0.967	0.012	0.005

Variable(s) entered in step 1: Sex, Age, Work experience, Teacher Efficacy, School Connectedness, Total PSQI, TOTAL_HPLPQ. y: years.

## Data Availability

The data presented in this study are available on request from the corresponding author. The data are not publicly available due to privacy.

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
