# Peer review of "Stress and Well-Being of Greek Primary School Educators: A Cross-Sectional Study"

_ijerph, 2023, doi:10.3390/ijerph20075390_

Round 1

Reviewer 1 Report

Spell check it again.

Author Response

No comments from the reviewer 1.

Reviewer 2 Report

This study demonstrated the relevance of stress in teachers. However, the method of study was not proper. Therefore, the results were questionable.

For example:

1. Subheading 2.2.1 teachers’ testing. Were 786 participants involved in the questionnaire testing the same participants involved in the actual study? Why were so many participants needed for the testing phase? Please explain. The participants should be different persons for the testing and actual study.

2. A synchronic descriptive epidemiologic study. What did the authors mean by “synchronic”? Is it referring to cross-sectional or spatial? I encourage the authors to use a standard term for better understanding.

3. Was the sample size calculated before the study? Please explain how the authors know the sample size was enough for the study to achieve an acceptable power of study.

4. Are there any inclusion and exclusion criteria for selecting the participants? Please include the criteria in the article.

5. The authors should briefly report the validity and reliability of the Greek translation questionnaire that was done in the test-pre-test procedure in this article.

6. Line 254. Why was the median used in the quantitative variables? Why was the interquartile range left out in the method section?

7. Why was the chi-square test used to correlate the non-categorical variables? Was the chi-square supposed to use for categorical data?

8. There was no description of logistic regression in the method section, but there was a result for the analysis in the result section.

Author Response

Response to Reviewer 2 Comments

Point 1: Subheading 2.2.1 teachers’ testing. Were 786 participants involved in the questionnaire testing the same participants involved in the actual study? Why were so many participants needed for the testing phase? Please explain. The participants should be different persons for the testing and actual study.

Response 1: From line 187-200 the procedure for the test-pre-test for the questionnaire is described. 786 participated in the validation of the questionnaire, and 40 individuals were involved in the testing of the questionnaire before it was distributed to the target population. These individuals were not included in the distribution for the final procedure.

There was incorrect formatting at this point, resulting in confusion for the reader. The heading 2.2.1 teachers' testing was removed from the article and there is continuity in the flow of the text.

Point 2: A synchronic descriptive epidemiologic study. What did the authors mean by “synchronic”? Is it referring to cross-sectional or spatial? I encourage the authors to use a standard term for better understanding.

Response 2: It refers to a cross-sectional study and will be corrected in the text.

Point 3: Was the sample size calculated before the study? Please explain how the authors know the sample size was enough for the study to achieve an acceptable power of study.

Response 3: It was calculated using Cochran's Formula and as we don't have much information on the subject to begin with, we assumed that half of the teachers have problems with stress, sleep and the other variables in the sample: this gives us the maximum possible variability. Therefore, p = 0.5. We wanted 95% confidence, and at least 5%-above or below precision. A 95% confidence level gives us Z values of 1.96, according to the normal tables, so we have ((1.96)2 (0.5) (0.5)) / (0.05)2 = 385; therefore a random sample of 385 teachers in our target population was considered sufficient to give us the confidence levels we need.

Point 4: Are there any inclusion and exclusion criteria for selecting the participants? Please include the criteria in the article.

Response 4: The criterion for inclusion was primary school teachers of any age in Greece and teachers of other levels were excluded.

Point 5: The authors should briefly report the validity and reliability of the Greek translation questionnaire that was done in the test-pre-test procedure in this article.

Response 5: Line 187-211. In addition, the article on the validation and reliability of the questionnaire for primary school teachers is in the process of being evaluated for publication.

Point 6: Line 254. Why was the median used in the quantitative variables? Why was the interquartile range left out in the method section?

Response 6: It was a clerical error and will be corrected and the interquartile range will be added. We used median, because the variables did not follow the Normal Distribution.

Point 7: Why was the chi-square test used to correlate the non-categorical variables? Was the chi-square supposed to use for categorical data?

Response 7: Line 258 it is inadvertently wrong and will be corrected.

Point 8: There was no description of logistic regression in the method section, but there was a result for the analysis in the result section.

Response 8: Logistic regression models were used to evaluate various participants’ characteristics as determinants of which category of perceived stress (Low/Moderate vs. High) each indi-vidual was likely to belong in

Reviewer 3 Report

The study has a cross-sectional design and is based on the self-administration of a questionnaire that investigates different dimensions through the use of structured and validated questionnaires.

The topic is of interest to the reader, the sample is large and the tools used are valid. in my opinion, the paper cannot be published in this form and requires major revision.

1)      Introduction

In this section it is necessary to clearly state the purpose of the research and the design. In fact, the study explores different dimensions and predictors but for what purpose? The collected data can in fact be used for various purposes.

2)      Materials and method

Regarding the "Participants and procedures" section: the authors should write something about the expected sample size with respect to the total potentially achievable population or a response rate.Also, in my opinion, it is not appropriate to describe results in the materials and methods section. The achieved sample and its main characteristics should be described in the results.

Regarding the "Measures" section and the "Teachers' testing" section: the reason for this division is not understood. The reason why 2 questionnaires are described in the first and the others in the second is not clear. I recommend doing one chapter. The important thing, in my opinion, is to carefully cite the validation works of the questionnaires to clarify whether they were validated in Greek or not.

Regarding the “Method of Data analysis” section: Quantitative variables should be described as mean and standard deviation or median and interquartile range based on the statistical distribution. This point is not clearly written and the choice seems arbitrary if not downright indifferent. In fact in table 2 both are reported as if they were alternatives and it is not correct. Furthermore, the first and third quartile of distribution should be reported as an interquartile range. The distribution of the linear variables must be investigated to direct the analysis towards a parametric type analysis or not. In this section both appear to have been done. In fact Spearman is used and after a linear regression that requires several assumptions to be performed. Are they satisfied? how were they analysed?

3)      Results

In this section, table 1 is correctly cited. In it, a univariate analysis of gender differences is performed. Was it a goal of the Authors? It was not made explicit either in the scope of the research or in the materials and methods. Please, the reader needs a clear document.

Table 2 shows, as I said earlier, some inappropriate results. It is necessary and useful to report only in terms of median and interquartile range (first and third quartile of distribution) the variables that have a statistically non-normal distribution and report only the mean and standard deviation for the variables with normal distribution.

Table 4 shows a linear regression between some predictors and the PSS results.

·       Was this analysis a purpose of the authors?

·       Why was only this questionnaire used as a dependent variable?

·       Are the assumptions for performing a linear regression satisfied?

·       How were the independent variables selected? Why only some?

all these considerations are proposed with the intention of improving the reading of an interesting work with a very large amount of data collected. If done constructively, I'm sure the paper will become more methodologically sound.

In the light of these revisions, it will also be appropriate to modify part of the discussions.

Major revisions are requested for the above reasons.

Author Response

Response to Reviewer 3 Comments

Point 1: 1)Introduction

In this section it is necessary to clearly state the purpose of the research and the design. In fact, the study explores different dimensions and predictors but for what purpose? The collected data can in fact be used for various purposes.

Response 1:

The following text was added ( line 139).

The area of teachers' well-being has not been sufficiently examined, as studies measuring subjective well-being are limited to the professional group of teachers. Teaching directly depends on teachers' well-being and is linked to teaching performance and is a key prerequisite for education. Teachers should feel satisfied with themselves to be able to function effectively in the teaching-learning process, to be able to provide favourable conditions for students in the educational context and to make the most of their ability to create appropriate conditions conducive to learning and the well-being of students in the educational context.

The need to measure teachers' well-being is considered an important element in understanding parameters such as the teacher's attachment to the school, self-efficacy in teaching and, ultimately, teachers' stress, elements that threaten their emotional and mental health and lead them to burnout syndrome.

Point 2: Results

Regarding the "Participants and procedures" section: the authors should write something about the expected sample size with respect to the total potentially achievable population or a response rate.Also, in my opinion, it is not appropriate to describe results in the materials and methods section. The achieved sample and its main characteristics should be described in the results.

Response : Because we had limited data regarding the topic to begin with, we assumed that half of the teachers have problems with stress, sleep, and the other variables in the sample: this gives us the maximum possible variability. As a result, p = 0.5. We desired 95% confidence and precision of at least 5% above or below. According to the normal tables, a 95% confidence level yields Z values of 1.96, so we have ((1.96)2 (0.5) (0.5)) / (0.05)2 = 385; thus, a random sample of 385 teachers in our target population was deemed sufficient to provide the confidence levels we require. The effects description has been removed.

Point :Regarding the "Measures" section and the "Teachers' testing" section: the reason for this division is not understood. The reason why 2 questionnaires are described in the first and the others in the second is not clear. I recommend doing one chapter. The important thing, in my opinion, is to carefully cite the validation works of the questionnaires to clarify whether they were validated in Greek or not.

Response : Line 187-211. In addition, the article on the validation and reliability of the questionnaire for primary school teachers is in the process of being evaluated for publication.

Line 220  … In Greece, Alexia Katsarou et al. performed the translation and reliability and validity testing of the Greek version of the PSS-14 [28] 

Line 224 the questionnaire has been created, formatted and validated in the Greek language, it will be mentioned in the text

Line 232 the questionnaire has been validated and tested for reliability in the Greek population by Kotronoulas GC, Papadopoulou CN, Papapetrou A, Patiraki E. [30], will also be reported in the text.

Point : Regarding the “Method of Data analysis” section: Quantitative variables should be described as mean and standard deviation or median and interquartile range based on the statistical distribution. This point is not clearly written and the choice seems arbitrary if not downright indifferent. In fact in table 2 both are reported as if they were alternatives and it is not correct. Furthermore, the first and third quartile of distribution should be reported as an interquartile range. The distribution of the linear variables must be investigated to direct the analysis towards a parametric type analysis or not. In this section both appear to have been done. In fact Spearman is used and after a linear regression that requires several assumptions to be performed. Are they satisfied? how were they analysed?

Response 3: Assumption 1: Linear Relationship

The points in the scatter plot roughly fall along a straight diagonal line, so there likely exists a linear relationship between the variables.

Assumption 2: No Multicollinearity

VIF values greater than 5* indicate potential multicollinearity. In our study the biggest VIF is 4,543

Assumption 3: Independence

Durbin-Watson test, if d is between 1.5 and 2.5 then autocorrelation is likely not a cause for concern. In our study d=1,879

Assumption 4: Homoscedasticity

The assumpion is met.

Assumption 5: Multivariate Normality

Multiple linear regression assumes that the residuals of the model are normally distributed. Q-Q plots show residuals that roughly follow a normal distribution.

Point 3: Results

In this section, table 1 is correctly cited. In it, a univariate analysis of gender differences is performed. Was it a goal of the Authors? It was not made explicit either in the scope of the research or in the materials and methods. Please, the reader needs a clear document.

Response : The following text is added to the paticipants and procedures section (line 152)

The primary aim of this study was to investigate stress and well-being of teachers. Well-being will be examined through the validation of the TSWQ questionnaire among Greek teachers.

Secondary aims of the study, were to examine gender differences and to look for predictors and correlates that may influence the complex issue of stress and well-being of teachers. It also aimed to extract data regarding healthy lifestyle adoption, sleep and demographic characteristics representative of the sample.

Point : Table 2 shows, as I said earlier, some inappropriate results. It is necessary and useful to report only in terms of median and interquartile range (first and third quartile of distribution) the variables that have a statistically non-normal distribution and report only the mean and standard deviation for the variables with normal distribution.

Response : We proceeded to correct the table and only variables with a statistically non-normal distribution are reported as median and interquartile range (first and third quartile of the distribution).

Point : Table 4 shows a linear regression between some predictors and the PSS results.

  • Was this analysis a purpose of the authors?
  • Why was only this questionnaire used as a dependent variable?
  • Are the assumptions for performing a linear regression satisfied?
  • How were the independent variables selected? Why only some?

Response :

  • Yes, it was within the aforementioned objectives of this study, as described in the objectives added (Line 155)
  • We proceeded to add the TSQW questionnaire for a broader perspective and drawing conclusions. Besides, the title of this study forewarns the reader of the contents to be analysed.
  • As mentioned in the previous question, all five conditions for conducting linear regression are met
  • The independent variables were selected according to the criterion of statistically significant correlation and for this reason the variable BMI was removed as it was not statistically significant